# Cell-Free DNA as a Biomarker at Diagnosis and Follow-Up in 256 B and T-Cell Lymphomas

**DOI:** 10.3390/cancers16020321

**Published:** 2024-01-11

**Authors:** Ramón Diez-Feijóo, Marcio Andrade-Campos, Joan Gibert, Blanca Sánchez-González, Lierni Fernández-Ibarrondo, Concepción Fernández-Rodríguez, Nieves Garcia-Gisbert, Laura Camacho, Marta Lafuente, Ivonne Vázquez, Luis Colomo, Antonio Salar, Beatriz Bellosillo

**Affiliations:** 1Department of Hematology, Hospital del Mar, 08003 Barcelona, Spain; rdiez-feijoo@psmar.cat (R.D.-F.); marcioandrade_hn@yahoo.com (M.A.-C.); bsanchezgonzalez@psmar.cat (B.S.-G.); 2Cancer Research Program, Hospital del Mar Research Institute, 08003 Barcelona, Spain; jgibert@psmar.cat (J.G.); lfernandez@researchmar.net (L.F.-I.); mconcepcionfernandezrodriguez@psmar.cat (C.F.-R.); nieves.garciagisbert@gmail.com (N.G.-G.); lcamacho@psmar.cat (L.C.); mlafuente@imim.es (M.L.); lcolomo@psmar.cat (L.C.); bbellosillo@psmar.cat (B.B.); 3Department of Pathology, Hospital del Mar, Hospital del Mar Research Institute, 08003 Barcelona, Spain; ivazquez@psmar.cat

**Keywords:** cfDNA, liquid biopsy, lymphoma, monitoring

## Abstract

**Simple Summary:**

A liquid biopsy is a minimally invasive strategy that provides useful information for the management of cancer patients. Cell-free DNA (cfDNA) extraction and quantification could be easily integrated into standard care protocols. This study analyzes cfDNA concentration as a biomarker in 256 lymphoma cases, including unexplored subtypes. The findings reveal that all lymphoma subtypes release higher cfDNA levels than controls, and these levels correlate with subtype, aggressiveness, and prognostic markers like the International Prognostic Index (IPI) and Ann Arbor Stage. Additionally, the study includes 49 large B-cell lymphomas (LBCL) cases, in which pre- and post-treatment levels of cfDNA were assessed, demonstrating that cfDNA dynamics correlate with treatment response and can, in some cases, improve PET-based response assessment. The study also includes NGS genetic characterization in tissue and cfDNA in a subgroup of LBCL. Overall, the study highlights cfDNA’s potential as a biomarker for lymphoma management, enabling its integration in the near future into routine clinical practice.

**Abstract:**

Background: Cell-free DNA (cfDNA) analysis has become a promising tool for the diagnosis, prognosis, and monitoring of lymphoma cases. Until now, research in this area has mainly focused on aggressive lymphomas, with scanty information from other lymphoma subtypes. Methods: We selected 256 patients diagnosed with lymphomas, including a large variety of B-cell and T-cell non-Hodgkin and Hodgkin lymphomas, and quantified cfDNA from plasma at the time of diagnosis. We further selected 49 large B-cell lymphomas (LBCL) and analyzed cfDNA levels at diagnosis (pre-therapy) and after therapy. In addition, we performed NGS on cfDNA and tissue in this cohort of LBCL. Results: Lymphoma patients showed a statistically significant higher cfDNA concentration than healthy controls (mean 53.0 ng/mL vs. 5.6 ng/mL, *p* < 0.001). The cfDNA concentration was correlated with lymphoma subtype, lactate dehydrogenase, the International Prognostic Index (IPI) score, Ann Arbor (AA), and B-symptoms. In 49 LBCL cases, the cfDNA concentration decreased after therapy in cases who achieved complete response (CR) and increased in non-responders. The median cfDNA at diagnosis of patients who achieved CR and later relapsed was higher (81.5 ng/mL) compared with levels of those who did not (38.6 ng/mL). A concordance of 84% was observed between NGS results in tumor and cfDNA samples. Higher VAF in cfDNA is correlated with advanced stage and bulky disease. Conclusions: cfDNA analysis can be easily performed in almost all lymphoma cases. The cfDNA concentration correlated with the characteristics of the aggressiveness of the lymphomas and, in LBCL, with the response achieved after therapy. These results support the utility of cfDNA analysis as a complementary tool in the management of lymphoma patients.

## 1. Introduction

Cell-free DNA (cfDNA) are short double-stranded DNA fragments (~166 base pairs) present in the circulation as a result of their release following cell death (by apoptosis, necrosis, or pyroptosis) and active cell secretion as microvesicles [1,2,3]. The amount of cfDNA in plasma is variable among individuals and may be modified by physiological situations like physical exercise or pregnancy, as well as chronic conditions such as cirrhosis or renal dysfunction, although several studies have reported that cfDNA mainly originates from hematopoietic cells [4,5,6,7]. One of the main interests in the analysis of cfDNA is focused on the fact that cancer patients have higher amounts of cfDNA than healthy subjects and that levels of cfDNA may change after exposure to radiotherapy or chemotherapy [8,9,10]. The cfDNA derived from cancer cells is known as circulating tumoral DNA (ctDNA). The opportunity to analyze the cfDNA present in the liquid compartments of the organism has generated the concept of liquid biopsy and has opened up an opportunity to improve the diagnosis and management of different cancer types. 

In the lymphoma setting, the value of cfDNA has been mainly studied in diffuse large B-cell lymphoma (DLBCL) [11,12,13] and Hodgkin lymphoma [14,15], but there is limited information in other lymphoma subtypes at the time of diagnosis.

Moreover, cfDNA is being postulated as an opportunity to monitor minimal residual disease and detect treatment failure or resistance [16,17]. In DLBCL, the study of cfDNA in plasma allows the identification of patients at risk of relapse 3.5 months before clinical evidence of disease, suggesting that cfDNA is a useful biomarker in this setting [11].

The aim of this work was to assess the amount of cfDNA at diagnosis in a cohort of 256 patients with several lymphoma entities from a single institution. Moreover, we carried out a substudy in a cohort of 49 LBCL patients to analyze the molecular profile by next-generation sequencing (NGS) in tissue and in liquid biopsy, as well as the kinetics of cfDNA levels after first-line therapy.

## 2. Materials and Methods

### 2.1. Patient Selection and Study Design

This is a retrospective single-center study including patients diagnosed with any subtype of lymphoma between January 2015 and March 2019 at Hospital del Mar. The study was carried out in accordance with the Declaration of Helsinki and the International Conference on Harmonization Guidelines for Good Clinical Practice and was approved by the local ethics committee. Patients gave informed consent for blood sample collection at diagnosis and during follow-up.

The diagnosis was conducted by histopathological and immunophenotypical studies of lymph nodes, extranodal tumors, and/or trephine bone marrow biopsies, according to the WHO revised 4th edition [18]. 

In this study, 256 lymphoma cases were included: 88 (34.8%) large B-cell lymphomas (LBCL) (60 diffuse LBCL not otherwise specified (DLBCL-NOS), 13 transformations from indolent B-cell lymphoma (tDLBCL), 7 primary mediastinal LBCL (PML), 5 primary LBCL of the central nervous system (DLBCL-CNS), and 3 high-grade B-cell lymphomas with *MYC* and *BCL2* rearrangement (HGBL)); 48 (18.8%) follicular lymphomas (FL); 35 (13.7%) marginal zone lymphomas (MZL) (21 extranodal MZL, 11 splenic MZL, and 3 nodal MZL); 31 (12.1%) classic Hodgkin lymphomas (cHL); 14 (5.5%) mantle cell lymphomas (MCL); 7 (2.7%) small lymphocytic lymphoma/chronic lymphocytic leukemia (SLL/CLL); 5 (2.0%) lymphoplasmacytic lymphoma/Waldenström macroglobulinemia (LpL/WM); 1 (0.4%) Burkitt lymphoma; 6 (2.3%) circulating low-grade unclassifiable B-cell lymphomas (LPS-NOS); 9 (3.5%) angioimmunoblastic T-cell lymphoma (AITL); 6 other T-cell lymphomas (2.3%; 2 anaplastic T-cell lymphomas of the skin, 1 primary cutaneous gamma/delta T-cell lymphoma, 1 peripheral T-cell lymphoma not otherwise specified, 1 transformed mycosis fungoides, and 1 T-cell prolymphocytic leukemia); and 6 others (2.3%; 5 HIV positive B-cell lymphomas, and 1 interdigitating dendritic cell sarcoma). In addition, 33 healthy subject controls were also included.

### 2.2. Sample Collection, cfDNA Isolation, and Quantification

After every patient signed the informed consent form and before the start of any kind of therapy, 10 mL of whole blood was collected in EDTA tubes. Within the first 4 h after extraction, the tubes were centrifuged two times at 2800 rpm for 10 min. Plasma was collected and used for immediate extraction of cfDNA or stored at −80 °C.

cfDNA isolation was performed with the MagMax Cell-free DNA Isolation Kit (Thermo Fisher Scientific, Foster City, USA) following the recommendations of the manufacturer. cfDNA was quantified using Qubit (Thermo Fisher Scientific, Waltham, MA, USA) and analyzed by electrophoresis to discard the presence of genomic DNA (4200 TapeStation system, Agilent, Santa Clara, CA, USA). 

### 2.3. Next-Generation Sequencing (NGS)

Molecular characterization was performed on tissue and cfDNA by NGS using 10–40 ng of cfDNA or 120 ng of DNA from FFPE samples. Libraries were prepared using a custom panel that covered the whole codifying region of 36 lymphoid-associated genes (QIAseq Custom DNA Panels, Qiagen, Hilden, Germany): *ARID1A*, *ATP6AP1*, *B2M*, *BCL2*, *BCL7A*, *CARD11*, *CCND3*, *CD58*, *CD79B*, *CDKN2A*, *CREBBP*, *EBF1*, *EP300*, *EZH2*, *FAS*, *FOXO1*, *GNA13*, *HIST1H1E*, *ID3*, *IDH2*, *IGLL5*, *KMT2D*, *MEF2B*, *MYD88*, *PIM1*, *PRDM1*, *RHOA*, *RRAGG*, *SGK1*, *SOCS1*, *STAT6*, *TCF3*, *TET2*, *TNFAIP3*, *TNFRSF14*, and *TP53*. Library preparation incorporated unique molecular identifiers (UMIs) to tag individual DNA molecules, which enables high-confidence variant detection by reducing false positives, PCR artifacts, and library bias. This library preparation system is based on primer extension and allows analysis of short-sized fragments of cfDNA. Libraries were sequenced with 2 × 150-bp paired-end reads using NextSeq (Illumina, San Diego, CA, USA) with a 3000× minimum read depth for tissue and 6000× for plasma. 

Sequencing files were processed using the GeneGlobe Data Analysis Center (Qiagen, Hilden, Germany) for FASTQ trimming, alignment to the reference genome, and generation of variant calling files (.vcf) (smCounter2, Qiagen, Hilden, Germany). The obtained variants were then annotated and classified using Illumina VariantStudio 3.0 software according to genomic databases (GenomAD, Varsome, cBioPortal, dbSNP, COSMIC, My Cancer Genome, and Cancer Genome Interpreter) and evidence of pathogenicity in the literature. Only variants classified as pathogenic or likely pathogenic were included in this study. The limit of detection established for variant detection was 5% variant allele frequency (VAF) in FFPE samples and 2% in cfDNA. Variants detected in the tissue were then sought in the paired plasma sample. In cases with low VAF, variants were confirmed visually using the Integrative Genomics Viewer (IGV) v2.8.9 software.

### 2.4. Statistical Analysis

An analysis of the amount of cfDNA was performed for the entire cohort and correlated with the major clinical characteristics. For the description and analysis of categorical variables in descriptive statistics, Pearson’s X2 or Fisher’s exact tests were used. For continuous variables, the Mann–Whitney U test and Kruskal–Wallis test were used. The overall survival (OS) was established from the date of diagnosis until the date of the last visit or death; to calculate the progression-free survival (PFS), the date of starting therapy and the date of confirmed progression or relapse were used. Survival curves were estimated using the Kaplan–Meier method with a 95% interval of confidence (IC). A subanalysis focused on LBCL patients, with samples taken at diagnosis and after therapy with R-CHOP/anthracyclin-containing immunochemotherapies was conducted. The response assessment in LBCL was performed using a PET-CT scan according to Lugano criteria. The statistical significance of the analysis was determined by the long-rank method, and two-tailed *p*-values of 0.05 or less were considered significant. All the analysis was performed using the SPSS V20.0 software package (Chicago, IL, USA) and R (RStudio Version 1.4.1106). 

## 3. Results

### 3.1. Patient Characteristics and Levels of cfDNA According to Lymphoma Subtype

From the 256 patients included in the study, cfDNA was quantifiable after extraction from 1 mL of plasma in 249 (97.3%) cases. In 7 cases, cfDNA presence was undetectable by fluorometric analysis, which were mainly localized lymphomas according to Ann Arbor (AA) stage: 1 cHL (stage II), 5 MZL (1 nodal, stage II; 4 MALT: 1 gastric, stage II; 1 intestinal, stage I; and 1 conjunctival, stage I and 1 primary cutaneous, stage I), and 1 FL grade 2 (stage III). The main clinical characteristics of the 249 cases with evaluable cfDNA are listed in Table 1. With respect to controls, 18 cases (55%) were females and 15 (45%) were males, with a median age of 59 (51–69) years. 

The median concentration of cfDNA obtained from healthy controls and cases according to lymphoma subtypes is shown in Table 2 and Figure 1. The median cfDNA concentration obtained from plasma was 27.8 ng/mL (IQR 14.5–56.0) for the lymphoma cases and 5.2 ng/mL (IQR 2.8–7.1) for healthy controls. There were no statistically significant differences in the levels of cfDNA among the cases according to age (<60 years vs. >60 years), gender, or bone marrow involvement. However, significantly higher levels of cfDNA were found in lymphoma patients with elevated levels of lactate dehydrogenase (LDH), elevated levels of beta 2-microglobulin (B2-MG), and advanced stage or B-symptoms (Appendix A). Moreover, levels of cfDNA were significantly correlated with LDH (r = 0.320, *p* < 0.0001), B2-MG (r = 0.253, *p* < 0.0001), and age (r = 0.157, *p* = 0.013) (Spearman test, Appendix A).

The median cfDNA concentration was significantly higher in all lymphoma categories when compared to healthy controls (Appendix A). Aggressive NHL such as LBCL, MCL, and other TL showed the highest median amount of cfDNA, followed by AITL, MZL, cHL, and FL (Table 2). 

As shown in Figure 2A, analysis of cfDNA levels across LBCL subtypes showed that they were particularly elevated in patients with high-grade lymphoma with rearrangement of *MYC* and *BCL2* (median 518 ng/mL) in comparison with patients with other LBCL subtypes (*p* < 0.001). DLBCL-NOS and tDLBCL had similar levels of cfDNA, both being about the double of the levels of patients with PML or primary DLBCL-CNS (48.5 ng/mL and 54 ng/mL vs. 24.8 ng/mL and 29.4 ng/mL, respectively, *p* = 0.057). One patient with early-stage BL localized in the intestine also had detectable levels of cfDNA.

Levels of cfDNA across MZL subtypes are shown in Figure 2B. In MZL, the median cfDNA concentration was 20.8 ng/mL (IQR: 14.2–29.3). Patients with splenic MZL had higher levels of cfDNA than extranodal MZL (*p* = 0.028) and nodal MZL (*p* = 0.231).

Of note, one patient with interdigitating dendritic cell sarcoma showed high cfDNA levels (47 ng/mL), and five patients with HIV+ B-cell lymphoma had median levels of 23.8 ng/mL (IQR: 2.6–37.3).

### 3.2. cfDNA Levels According to Characteristics at Presentation in the Main Lymphoma Subtypes

We further analyzed the levels of cfDNA according to the characteristics at presentation in the main lymphoma subtypes. In LBCL, levels of cfDNA were significantly higher in patients with B-symptoms (*p* = 0.001), high LDH (*p* = 0.003), high B2-MG (*p* = 0.002), and a higher International Prognostic Index (IPI) score (*p* = 0.006) (Figure 3). In FL, higher cfDNA levels were found in older patients (*p* = 0.017) and in those with advanced IPI (*p* = 0.012). In cHL and in MZL, cfDNA concentration was significantly correlated with age and LDH, respectively (Appendix A).

### 3.3. Molecular Characterization at Diagnosis Using cfDNA and Tissue-DNA in LBCL (n = 49) 

We further analyzed the applicability of cfDNA for molecular profiling and longitudinal follow-up in the cohort of LBCL. To this aim, we included 49 patients with end-of-therapy evaluation and at least 12 months of follow-up (N = 49; 34 DLBCL-NOS, 8 tDLBCL, 4 PML, 1 DLBCL-CNS, 1 HGBL, and 1 HIV-positive LBCL). The median age of this cohort was 62 years (range: 20–86); 27 (55%) were males, 16 (33%) had B-symptoms, and 27 (55%) showed an upper normal limit of LDH concentration. The AA stages were: I in 8 patients (16%), II in 15 patients (31%), III in 3 patients (6%), and IV in 23 patients (47%). The IPI distribution was low-risk in 14 patients (29%), low-intermediate in 18 patients (37%), intermediate-high in 12 patients (24%), and high-risk in 5 patients (10%) (Appendix A). The median cfDNA concentration from the 48 cases with plasma at the time of diagnosis was 38.4 ng/mL (IQR: 17.2–59.9). Similar to the whole LBCL cohort, patients with intermediate/high or high IPI had significantly higher cfDNA levels in comparison with those with low/low-intermediate IPI (median 58.10 ng/mL (IQR: 24.9–90.8) vs. 27.40 ng/mL (IQR: 14.9–48.1), respectively; *p* = 0.015) (Appendix A). 

In 33 out of 48 (68.8%) patients, the amount of isolated cfDNA at baseline was enough to perform mutational analysis by NGS. Sequencing results were evaluable in 23 out of 33 cases (in 10 cases, median read depth and, therefore, sensitivity were too low to perform a reliable analysis), showing one or more gene mutations in 21 out of 23 (91.3%) cases. To confirm the tumoral origin of the gene mutations detected in plasma, we performed NGS on the tumoral FFPE samples. Of the 49 cases included, tumor tissue was exhausted in 2 cases, extracted DNA was insufficient to perform NGS in 10 cases, and NGS data were non-evaluable in 5 cases. Successful NGS results from FFPE tissue were obtained in 32 out of 49 initial cases (65.3%). In 29 out of 32 cases, one or more gene mutations were identified. The distribution of gene mutations in both tissue samples (N: 32) and cfDNA samples (N: 21) is shown in Figure 4. The median number of mutations per sample, for both tissue-DNA and cfDNA, was 5 (IQR: 2.7–7.0 and 3.0–6.0, respectively). In agreement with previous reports, the more frequently mutated genes in both tissue and cfDNA were *KMT2D*, *CREBBP*, *TP53*, *IGLL5*, *BCL2*, *EBF1*, *TET2*, *EZH2*, and *MEF2B*. 

The paired tissue–plasma analysis was possible in 16 cases with valuable NGS. The correlation between the mutations identified in tissue-DNA and in cfDNA is shown in Figure 5. In these cases, a total of 82 gene mutations were found in tissue samples, and 69 (84%) of them were detected in the cfDNA samples. When analyzing the variant allele frequency (VAF) of the mutations, the median VAF for tissue-DNA samples was 30.65% (IQR: 23.90–45.19%) and 13.77% for cfDNA samples (IQR: 4.77–32.79%). A correlation was observed between tissue-DNA VAF and cfDNA VAF (r = 0.426, *p* < 0.0001) (Figure 6A). cfDNA VAF was also correlated with cfDNA concentration (r = 0.484, *p* < 0.05). In the cases in which mutations were missed in cfDNA, the median VAF in tissue was significantly lower than cases in which mutations were detected in cfDNA (20.94% (IQR: 12.15–27.05%) vs. 32.20% (IQR: 24.66–48.10%), respectively; *p* < 0.0001).

We further analyzed whether these differences in the VAF could be related to the clinical presentation of the disease. To this aim, we defined two groups: group A, which included patients with AA stages I or II and the absence of bulky disease, and group B, which comprised patients with advanced stages and/or bulky disease. Group A exhibited statistically significantly lower VAF percentages than group B in cfDNA samples (*p* = 0.0013) (Figure 6B).

### 3.4. cfDNA Concentration as a Prognosis Biomarker before and after Therapy for LBCL (n = 49)

To explore the applicability of cfDNA to monitor disease, we compared the amount of cfDNA obtained at diagnosis and at the end of therapy. All 49 LBCL patients received immunochemotherapy regimens with curative intention, 37 cases (76%) received R-CHOP (rituximab, cyclophosphamide, doxorubicin, vincristine, and prednisone), 8 patients (16%) received R-EPOCH-DA (rituximab, etoposide, prednisone, vincristine, doxorubicin, and cyclophosphamide), and the remaining 4 patients (8%) received other rituximab-containing regimens. After therapy completion, response evaluation following the Lugano criteria was as follows: complete response (CR) in 42 patients (85.7%), partial response (PR) in 1 (2%), stable disease (SD), or disease progression (PD) in 6 (12.2%).

As stated above, at diagnosis, cfDNA isolation was possible in 48 patients (from 1 patient, there was not enough plasma sample at diagnosis) with a median cfDNA concentration of 35.5 ng/mL (IQR: 17.2–58.1); after therapy, cfDNA isolation could be done in 43 patients (5 patients did not have a sample after treatment), and the median cfDNA concentration was 24.2 ng/mL (IQR: 13.6–35.8). Figure 7 summarizes the levels of the cfDNA before and after therapy according to response. Patients who achieved response showed a reduction of median levels of cfDNA from diagnosis to response evaluation (from 42.7 ng/mL to 22.0 ng/mL, respectively) (*p* = 0.007). However, median levels of cfDNA increased in patients who did not achieve response (SD/PD) (from 18.1 ng/mL to 57.6 ng/mL (*p* = 0.08)). 

With a median follow-up for survivors of 50.9 months (IQR: 40.7–66.6), 4 patients relapsed at 13, 19, 25, and 41 months. Three out of these four cases achieved a second remission after treatment. The median cfDNA at diagnosis of patients who achieved CR and later relapsed was higher (median: 81.5 ng/mL; IQR: 57.0–124.5) compared with levels of those in CR who did not relapse (median: 38.6 ng/mL (IQR: 17.4–50.9)). At the end of treatment (EOT), the median cfDNA was not different between those cases in CR who relapsed and those who did not (median: 22.0 ng/mL (IQR: 20.0–24.4) vs. 24.2 ng/mL (IQR: 11.7–36.1), respectively).

With this follow-up, eight patients died, and the median OS has not been reached. Five of these patients had a positive PET at EOT, and, interestingly, in three of them, the cfDNA had increased with respect to cfDNA levels at diagnosis. In all five cases, the main cause of death was lymphoma progression. The three remaining patients who died had achieved CR with negative PET and showed diminished levels of cfDNA at EOT. Conversely, in only one of these three cases, the cause of death was lymphoma progression. 

We further explored the usefulness of adding the pre- and post-treatment cfDNA kinetics to EOT-PET. At EOT, seven patients were PET+ (three patients showed a reduction in cfDNA levels, three patients showed an increase in cfDNA, and one patient showed no significant cfDNA variation). Thirty-six patients were PET− at EOT (25 patients showed a reduction in cfDNA levels, 8 showed an increase in cfDNA, and 3 patients showed no significant cfDNA variation). Figure 8 illustrates PFS and OS according to the combined PET response at EOT to cfDNA kinetics. As expected, patients who achieved metabolic CR (Deauville Score (DS) 1–3) had prolonged PFS and OS, regardless of final cfDNA kinetics. However, the kinetics of cfDNA at EOT was able to distinguish those patients who, despite being considered PET positive (DS 4–5), showed a favorable outcome. Thus, the PFS and OS of patients with PET positivity (DS 4–5) who had decreased levels of cfDNA at EOT were similar to the outcomes of those PET-negative patients. Of note, PET-positive (DS 4–5) patients who had increased cfDNA kinetics at EOT showed a very poor outcome. 

In the patients with ctDNA available, levels of ctDNA at the time of diagnosis were measured in haploid genome equivalents per milliliter (hGE/mL) and expressed on the log scale (log hGE/mL). Using the threshold of 2.5 log hGE/mL, as previously described [12], differences in PFS and OS were observed, but without reaching statistical significance, probably due to the small sample size. None of the three cases below 2.5 log hGE relapsed (Appendix A).

## 4. Discussion

In this single-center study, we have analyzed cfDNA in 256 patients with lymphoma, including the most frequent non-Hodgkin lymphoma entities, Hodgkin lymphoma, and also less common lymphomas, such as LpL/WM or AITL. cfDNA was successfully isolated in up to 97% of our patients. The cases in which cfDNA could not be detected corresponded to patients with localized stages and/or with very little tumor burden and, mostly, indolent lymphomas (in particular, MZL). 

Overall, cfDNA concentration was higher in all lymphoma categories, including indolent lymphomas, compared to healthy controls. Furthermore, the median concentration of cfDNA was higher in patients with aggressive lymphoma. Differences were also seen regarding the LBCL subtype, with greater cfDNA levels seen in HGBL, DLBCL-NOS, and tDLBCL than in DLBCL-CNS and PML. In particular, patients with HGBL with *MYC* and *BCL2* rearrangements presented strikingly higher cfDNA levels (10 times higher) than other categories of aggressive LBCL.

An interesting observation of our study was that although the MZLs had only a cfDNA isolation rate of 86%, the average concentration was approximately 20.8 ng/mL, sufficient enough to perform ctDNA analysis and higher than that reported in the recent MAGNOLIA study, which was 6.9 ng/mL [19]. However, cfDNA levels depend on multiple technical factors, including sample processing and nucleic acid extraction methods, which are not yet standardized, and therefore it is difficult to make comparisons among studies analyzing cfDNA [20,21]. 

Our study showed that baseline cfDNA levels were significantly correlated with stage, B-symptoms, as well as LDH and B2-MG concentrations. Interestingly, older patients also had higher levels of cfDNA. In our cohort of patients with LBCL, cfDNA concentrations at diagnosis were correlated with IPI, LDH, and B2-MG, as previously described by other groups that analyzed ctDNA [11,15,22]. In FL, cfDNA levels are correlated with age and IPI. The relationship between circulating tumor DNA (ctDNA) levels and tumor burden or risk scores in FL remains incompletely defined. Nevertheless, recent studies have established correlations between baseline ctDNA, advanced disease stages, FLIPI score, and prognosis [23]. In cases of progressive FL with histological transformation, ctDNA has demonstrated the capability to detect gene mutations related to transformation that are not discernible in tumor biopsies [24]. In HL, both the amount of cfDNA and ctDNA concentrations at baseline have been correlated with stage, mutational tumor burden, and the International Prognostic Score (IPS) [25]. In our study, cfDNA concentration was only significantly correlated with age and LDH. Camus et al. also reported higher pre-treatment cfDNA levels in patients >45 years in a series of 60 cHL [26].

Several studies have highlighted the opportunity of using cfDNA as a source of genetic material not only for molecular study of aggressive lymphomas, especially DLBCL [11,12,27,28,29,30,31,32], but also in HL [14,26,33,34], FL [35,36,37], MCL [38,39], TCL [40,41], or PML [42,43]. Non-invasive genotyping by ctDNA profiling from blood plasma might represent a complementary tool or, in the near future, an alternative option for lymphoma diagnosis, subtype classification, and genetic characterization. These aspects become particularly crucial in scenarios where accessing tumor tissue is challenging or in patients with a high risk of complications due to comorbidities. Additionally, ctDNA profiling can reveal tumor heterogeneity originating from various tumoral sites.

In our LBCL cohort, genetic characterization both in tissue and plasma was performed by the NGS-customized panel, including 36 genes, which was able to identify at least one or more mutations in 90% of cases with available material. Interestingly, the median number of mutations per sample for both tissue-DNA and cfDNA was the same (5 mutated genes). Matched DNA-tissue and cfDNA analysis helped us to confirm the tumoral origin of the cfDNA and to distinguish clonal hematopoiesis-related gene mutations. The findings from the genetic characterization in our LBCL cohort, despite the small sample size, were in line with previous studies [32,44,45]. Of note, genes related to prognosis and susceptible to targeted therapy, such as *KMT2D*, *TP53*, *CREBBP*, and *EZH2*, were identified in ctDNA by our panel. 

The genetic characterization of LBCL in cfDNA showed 84% concordance versus DNA-tissue. cfDNA generally exhibited lower VAF in comparison to the corresponding DNA-tissue sample, and a higher detection rate in ctDNA was associated with a higher median VAF in tissue (around 30%). We hypothesized that these low VAF tissue mutations missed in the cfDNA analysis could have a subclonal origin. These findings align with previous reports across various lymphoma subtypes [46]. Moreover, LBCL patients with advanced Ann Arbor stage (III–IV), and those with bulky disease showed significantly higher VAF in ctDNA than those with localized disease or absence of bulky disease. This finding is in line with other solid tumors for which higher VAFs have been detected in more advanced and aggressive disease [47,48], probably due to a higher ctDNA/cfDNA ratio. Our deep sequencing NGS panel incorporates the tagging of individual DNA molecules, facilitating ctDNA detection at low levels. However, it is well known that in cases with extremely low tumor burden or during or after treatment in responding patients, current methods are limited by suboptimal sensitivity. The development of other approaches, such as bioinformatic strategies like ‘PhasED-seq’ (phased variant enrichment and detection sequencing), which tracks two or more variants (‘phased variants’) on the same strand of one single DNA molecule, may increase the sensitivity of variant detection, thus facilitating ctDNA monitoring down to an analytical detection limit of ~0.00005% [46,49].

The utility of baseline ctDNA concentration has been investigated for predicting treatment outcomes. Additionally, the measurement of ctDNA during treatment, at the conclusion of therapy, and throughout surveillance has also been explored. A higher concentration of cfDNA at diagnosis has been associated with poor survival in a study including 40 DLBCL [28]. Several studies that evaluated pre-treatment ctDNA concentrations showed strong predictive value for PFS and OS in patients receiving standard immunochemotherapy [11,12,29,50] and also in patients receiving CART (Axi-cel) [51]. In our study, ctDNA levels at baseline were not found to have significance in the outcome after immunochemotherapy, probably due to the fact that our cohort of LBCL had to have end-of-therapy evaluation and also at least 12 months of follow-up, and consequently, progressing patients during treatment were not included in the cfDNA subanalysis. However, our patients who achieved CR but then relapsed had significantly higher cfDNA levels than those patients who also achieved CR but did not relapse.

As expected, in responding patients, cfDNA concentration decreased from diagnosis to the end of treatment. However, the cfDNA concentration increased in patients who showed progression at the end of treatment. In fact, a reduction of 2.5 log hGE/mL from diagnosis to the end of treatment showed differences in PFS and OS, but without reaching statistical significance, probably due to the limited number of events. The ctDNA dynamics is something that has been recently explored in some lymphoma studies and clinical trials to predict outcome. Kurtz et al. observed in their 217 DLBCL series that a 2-log reduction in ctDNA after one cycle of therapy and 2.5-log reductions after two cycles were highly predictive of EFS and OS after first-line therapy [12]. Similarly, negative ctDNA levels after CART therapy were associated with a durable response in a series of DLBCL patients [51,52]. Another phase 1 trial for recurrent/refractory CNS lymphomas treated with ibrutinib demonstrated a better outcome for those patients who cleared ctDNA in cerebrospinal fluid after therapy [53].

Finally, we studied the value of cfDNA with respect to the metabolic response evaluated by PET. In line with previous studies, in patients with negative PET (DS 1–3) at the end of treatment, the cfDNA concentration did not show predictive usefulness in the outcome. However, patients with positive PET (Deauville score 4–5) and a reduction in cfDNA at the end of treatment had a favorable prognosis, unlike those with positive PET and an increase in cfDNA, who had a dismal outcome [54,55]. 

## 5. Conclusions

In conclusion, cfDNA can be isolated in more than 95% of lymphoma patients in routine clinical practice, including B-cell, T-cell, and Hodgkin lymphomas, as well as aggressive and indolent lymphomas. cfDNA concentration at diagnosis varies among lymphoma subtypes, showing higher levels in more aggressive lymphoma types. In LBCL, the NGS-customized panel on cfDNA allowed the identification of 84% of the mutated genes detected in tissue-DNA. Moreover, cfDNA might help with PET response evaluation in LBCL. Our results add evidence for the incorporation of cfDNA/ctDNA measurement as a complementary tool for lymphoma management.

## Figures and Tables

**Figure 1 cancers-16-00321-f001:**
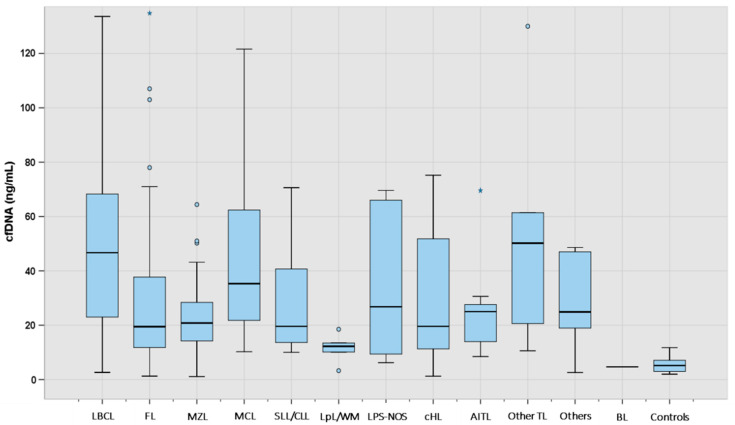
cfDNA levels at diagnosis according to lymphoma subtype (N: 249). The “box” represents the middle 50% of the data, with the top edge of the box indicating the third quartile (75th percentile) and the bottom edge indicating the first quartile (25th percentile). The median is represented inside each box. The “lines” (whiskers) extend from the box to show the range of the data (1.5 times IQR), excluding outliers that are represented with separated blue dots or stars. LBCL: large B-cell lymphoma; FL: follicular lymphoma; MZL: marginal zone lymphoma; MCL: mantle cell lymphoma; SLL/CLL: small lymphocytic lymphoma/chronic lymphocytic leukemia; LpL/WM: lymphoplasmacytic lymphoma/Waldenström macroglobulinemia; LPS-NOS: circulating low-grade unclassifiable B-cell lymphoma; cHL: classic Hodgkin lymphoma; AITL: angioimmunoblastic T-cell lymphoma; TL: T-cell lymphoma; BL: Burkitt lymphoma.

**Figure 2 cancers-16-00321-f002:**
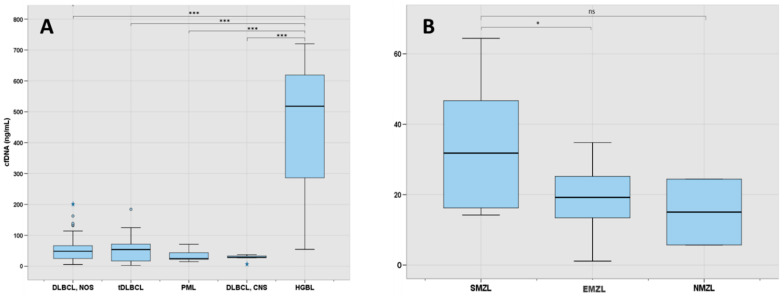
(**A**) cfDNA levels at diagnosis in the LBCL subgroup (N 88). (**B**) cfDNA at diagnosis in the MZL subgroup (N:30). Here, 1 asterisk (*) indicates a *p*-value of <0.05, 3 asterisks (***) indicate a *p*-value of <0.001 and ns indicates no statistical significance. The “lines” (whiskers) extend from the box to show the range of the data (1.5 times IQR), excluding outliers that are represented with separated blue dots or stars. DLBCL: diffuse large B-cell lymphoma; NOS: not otherwise specified; tDLBCL: transformed large cell lymphoma; PML: primary mediastinal lymphoma; CNS: central nervous system; HGBL: high-grade B-cell lymphoma with *MYC* and *BCL2* rearrangement; SMZL: splenic marginal zone lymphoma; EMZL: extranodal marginal zone lymphoma; NMZL: nodal marginal zone lymphoma.

**Figure 3 cancers-16-00321-f003:**
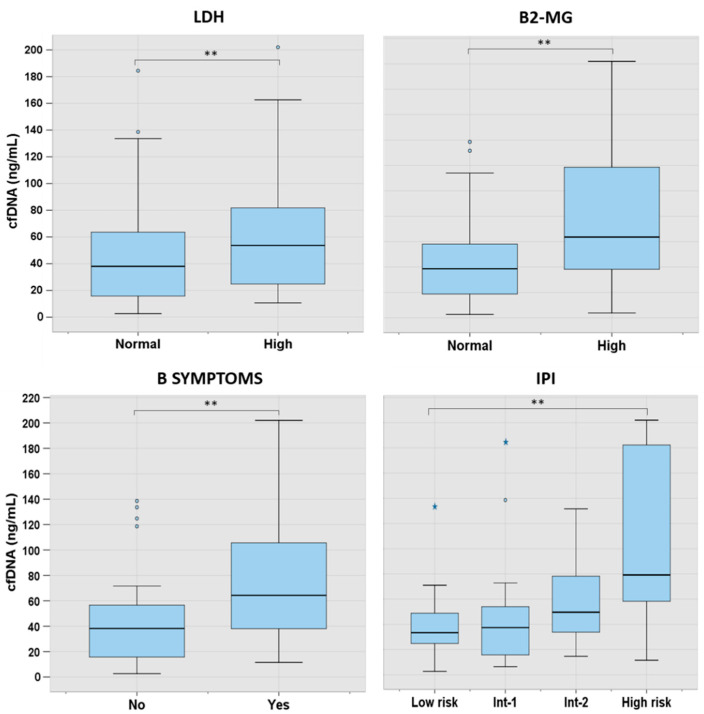
cfDNA levels according to characteristics at presentation in LBCL. The “lines” (whiskers) extend from the box to show the range of the data (1.5 times IQR), excluding outliers that are represented with separated blue dots or stars. Here, 2 asterisks (**) indicate a *p*-value of <0.01. Int: intermediate.

**Figure 4 cancers-16-00321-f004:**
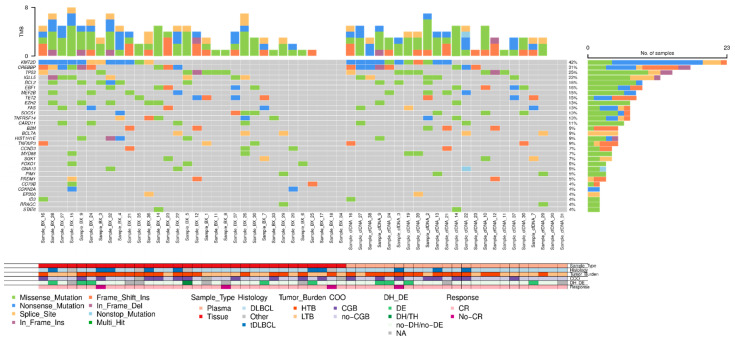
Mutational profile of LBCL patients assessed in tissue-DNA (N = 32) and cfDNA (N = 23). DLBCL: diffuse large B-cell lymphoma, tDLBCL: transformations of indolent B-cell lymphoma, HTB: high tumor burden, LTB: low tumor burden, CR: complete response, DE: double expressor, DH: double hit, TH: triple hit, NA: not available.

**Figure 5 cancers-16-00321-f005:**
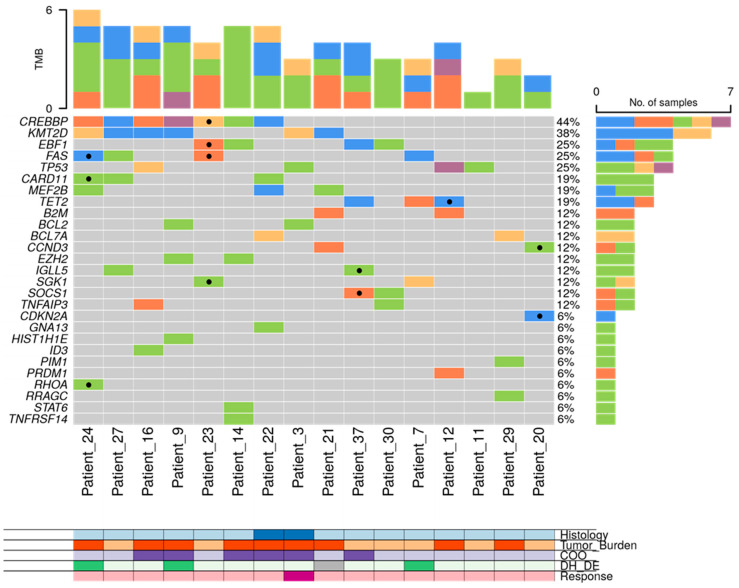
Analysis of the mutational profile in 16 DLBCL with paired tissue and cfDNA samples. The black dots indicate the mutations present in the tissue that were not detected in the cfDNA. DLBCL: diffuse large B-cell lymphoma, tDLBCL: transformations of indolent B-cell lymphoma, HTB: high tumor burden, LTB: low tumor burden, CR: complete response, DE: double expressor, DH: double hit, TH: triple hit, NA: not available.

**Figure 6 cancers-16-00321-f006:**
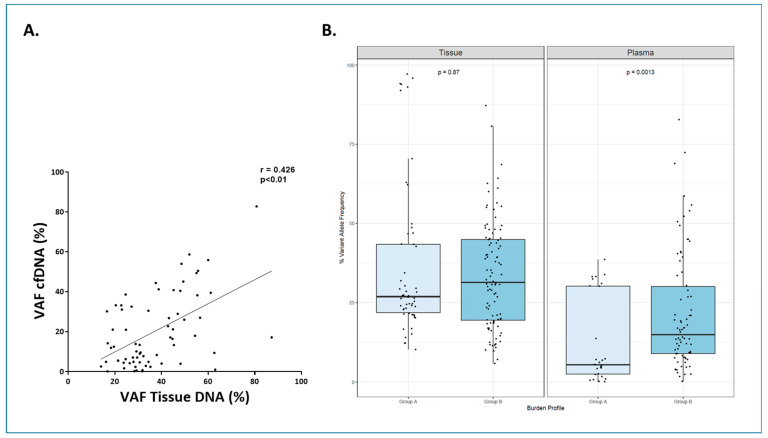
(**A**) Scattered plot showing correlation between variant allele frequency (VAF) in cfDNA and tissue-DNA. (Spearman test). (**B**) Correlation between variant allele frequency (VAF) in cfDNA and tissue-DNA and tumor burden in LBCL. Patients were classified in group A: Ann Arbor stage I/II and absence of bulky disease and group B: Ann Arbor stage III/IV or presence of bulky disease. In general, the variant allele frequency (VAF) of the mutations was higher in tissue-DNA independently of the group. When comparing the groups among them, group B vs. group A showed a statistically significantly higher VAF in cfDNA samples (*p* = 0.0013). The dots represent gene mutations. The “boxes” represent the middle 50% of the data, with the top edge of the boxes indicating the third quartile (75th percentile) and the bottom edge indicating the first quartile (25th percentile). The median is represented inside each box. The “lines” (whiskers) extend from the box to show the range of the data (1.5 times IQR).

**Figure 7 cancers-16-00321-f007:**
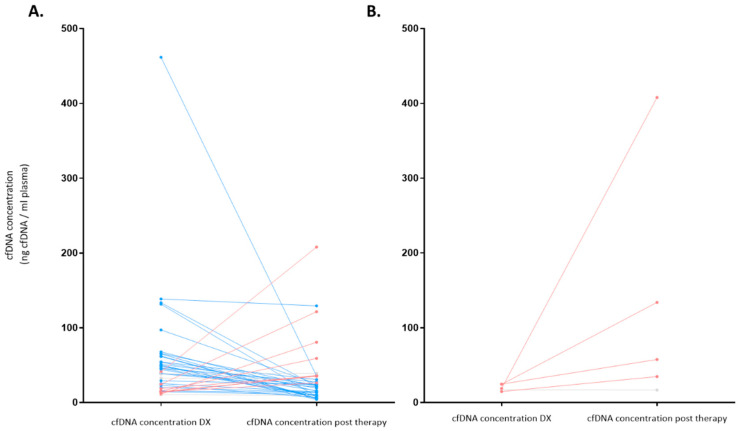
cfDNA concentration (ng/mL) before and after therapy according to response (N = 43). Significant variations in cfDNA concentration were considered when at least 10% variation was observed between pre- and post-treatment values. Significant decreases are colored blue, significant increases are colored red, and no variations are colored gray. (**A**) represents patients who achieved a complete or partial response at the end of treatment (EOT). (**B**) represents non-responder patients who had stable or progressive disease at EOT. DX: diagnosis.

**Figure 8 cancers-16-00321-f008:**
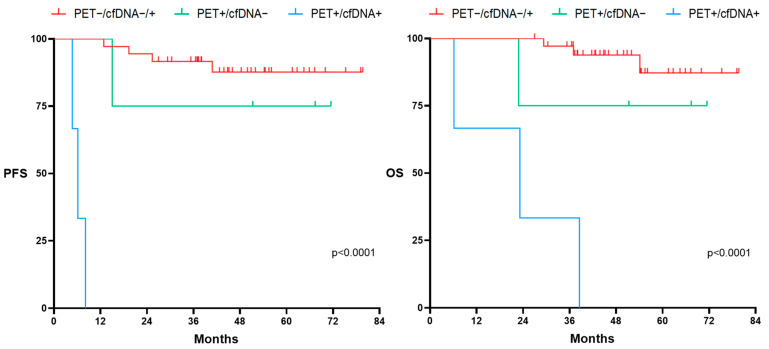
Progression-free survival (PFS) and overall survival (OS) according to PET at the end of treatment (EOT) and kinetics of cfDNA in the LBCL cohort (N:43). The PET−/cfDNA−/+ group encompasses patients with a negative PET at EOT, regardless of the cfDNA kinetics. PET+/cfDNA− includes patients with positive PET at the EOT and decreased levels of cfDNA at the EOT. PET+/cfDNA+ encompasses patients with positive PET at the EOT and increased levels of cfDNA at the EOT. Significant variations in cfDNA concentration were considered when at least 10% variation was observed between pre- and post-treatment values.

**Table 1 cancers-16-00321-t001:** Clinical features of 249 patients with evaluable cfDNA at diagnosis. AA: Ann Arbor; LDH: lactate dehydrogenase; BM: bone marrow. LBCL: large B-cell lymphoma; FL: follicular lymphoma; MZL: marginal zone lymphoma; MCL: mantle cell lymphoma; SLL/CLL: small lymphocytic lymphoma/chronic lymphocytic leukemia; LpL/WM: lymphoplasmacytic lymphoma/Waldenström macroglobulinemia; BL: Burkitt lymphoma; LPS-NOS: circulating low-grade unclassifiable B-cell lymphoma; cHL: classic Hodgkin lymphoma; AITL: angioimmunoblastic T-cell lymphoma; TL: T-cell lymphoma.

	LBCL (N = 88)	FL (N = 47)	MZL (N = 30)	MCL (N = 14)	SLL/CLL (N = 7)	LpL/WM (N = 5)	BL (N = 1)	LPS-NOS (N = 6)	cHL (N = 30)	AITL (N = 9)	Other TL (N = 6)	Others (N = 6)
Age, years (range)	67 (19–91)	62 (36–86)	71 (43–93)	71 (49–84)	77 (46–82)	70 (41–89)	34	81 (47–87)	40 (18–86)	62 (28–91)	59 (45–79)	48 (35–76)
Gender												
Male	58%	45%	50%	86%	71%	100%	–	17%	53%	67%	67%	100%
Female	42%	55%	50%	14%	29%	–	100%	83%	47%	33%	33%	–
AA stage												
I–II	43%	37%	57%	–	–	–	100%	–	53%	–	23%	17%
III–V	57%	63%	43%	100%	100%	100%	–	100%	47%	100%	77%	83%
B symptoms	41%	22%	10%	38%	25%	0	0	0%	47%	33%	17%	83%
LDH elevated	52%	11%	10%	36%	0	0	0	17%	7%	33%	40%	83%
BM involvement	15%	34%	30%	86%	100%	100%	0	31%	20%	22%	17%	17%

**Table 2 cancers-16-00321-t002:** cfDNA levels (ng/mL) according to lymphoma subtypes and controls. LBCL: large B-cell lymphoma; DLBCL: diffuse large B-cell lymphoma; NOS: not otherwise specified; PML: primary mediastinal lymphoma; CNS: central nervous system; HGBL: high-grade B-cell lymphoma with *MYC* and *BCL2* rearrangement; FL: follicular lymphoma; MZL: marginal zone lymphoma; NMZL: nodal marginal zone lymphoma; EMZL: extranodal marginal zone lymphoma; SMZL: splenic marginal zone lymphoma; MCL: mantle cell lymphoma; SLL/CLL: small lymphocytic lymphoma/chronic lymphocytic leukemia; LpL/WM: lymphoplasmacytic lymphoma/Waldenström macroglobulinemia; BL: Burkitt lymphoma; LPS-NOS: circulating low-grade unclassifiable B-cell lymphoma; cHL: classic Hodgkin lymphoma; AITL: angioimmunoblastic T-cell lymphoma; TL: T-cell lymphoma; IQR: interquartile range.

Type of Lymphoma	N	Median (ng/mL)	IQR (ng/mL)
LBCL	88	46.0	23.2–68.2
DLBCL, NOS	60	48.5	24.8–67.0
Transformed LBCL	13	54.0	15.9–95.1
PML	7	24.8	21.2–48.6
DLBCL, CNS	5	29.4	16.9–35.4
HGBL	3	518.0	286.3–619.0
FL	47	19.5	11.7–39.2
MZL	30	20.8	14.2–29.3
NMZL	2	15.1	5.7–24.4
EMZL	17	19.2	13.2–26.7
SMZL	11	31.8	15.8–50.2
MCL	14	35.3	19.5–77.2
SLL/CLL	7	19.6	11.5–53.6
LpL/WM	5	12.2	6.7–16.0
BL	1	4.68	NA
LPS-NOS	6	26.8	8.6–66.9
cHL	30	19.6	11.0–53.4
AITL	9	25.0	14.0–27.6
Other TL	6	50.2	26.3–6.3
Others	6	24.9	20.2–41.8
All cases	249	27.8	14.5–56.0
Controls	33	5.2	2.8–7.1

## Data Availability

The data that support the findings of this study are available from the corresponding author, A.S., upon reasonable request.

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
