# Peer review of "Cell-Free DNA as a Biomarker at Diagnosis and Follow-Up in 256 B and T-Cell Lymphomas"

_cancers, 2024, doi:10.3390/cancers16020321_

Round 1

Reviewer 1 Report

Comments and Suggestions for Authors

Summary of the main findings of the study. 

In this study, the authors analyzed cfDNA in 256 lymphoma cases. The main findings include higher cfDNA in lymphoma than in control, cfDNA levels correlate with aggressiveness, prognostic markers, In LBCL, cfDNA kinetics correlate with treatment response and may help to modify the PET-based response assessment. 

The strengths of this study 

Strengths: This study reveals that cfDNA has the potential to work as prognostic markers and be used to assess response to therapy for lymphoma. 

Comments

It’s interesting that in Figure 6B, Group A exhibited statistically significant lower VAF percentages than group B in cfDNA samples, but not in tissue-DNA samples while Supplementary Table 1 presented that higher levels of cfDNA was found in advanced stage. Did VAF percentages correlate with cfDNA level? Is higher VAF level an independent factor associated with advanced stage?

Author Response

RESPONSE TO REVIEWER 1:

Summary of the main findings of the study. 

In this study, the authors analyzed cfDNA in 256 lymphoma cases. The main findings include higher cfDNA in lymphoma than in control, cfDNA levels correlate with aggressiveness, prognostic markers, In LBCL, cfDNA kinetics correlate with treatment response and may help to modify the PET-based response assessment. 

We appreciate your time and effort in reviewing our manuscript. Enclosed are the specific responses and the appropriate modifications to the manuscript, which are highlighted in the resubmitted paper.

The strengths of this study 

Strengths: This study reveals that cfDNA has the potential to work as prognostic markers and be used to assess response to therapy for lymphoma. 

Comments

It’s interesting that in Figure 6B, Group A exhibited statistically significant lower VAF percentages than group B in cfDNA samples, but not in tissue-DNA samples while Supplementary Table 1 presented that higher levels of cfDNA was found in advanced stage. Did VAF percentages correlate with cfDNA level? Is higher VAF level an independent factor associated with advanced stage?

Thank you for this observation. In Figure 6B, only a cohort of LBCL patients with available cfDNA and tissue-DNA were included (n=49). In contrast, Supplementary Table 1 includes patients with plasma cfDNA concentration (ng/mL) regardless of lymphoma entity and therefore includes aggressive and indolent lymphomas of B and T-cell phenotype as well as Hodgkin lymphomas (n= 249). In the whole cohort, higher levels of cfDNA were found in advanced stage (25.5 vs 30.0 ng/ml, P=0.020). In the cohort of LBCL (n=49), cfDNA was not statistically associated with Ann Arbor stage, probably because the small number of cases. However, we found interesting to show that even VAF in cfDNA were correlated with VAF in tissue DNA (Figure 6A), those patients who had high tumor burden (defined as LBCL with advanced stage or those with localized stage but bulky disease) had a higher VAF than those who did not have. These data suggest that patients with higher VAF at liquid biopsy (higher ctDNA shedding) might have higher tumor burden according to our definition (“group B”) and might be of help to clinicians if confirmed in further studies. Therefore, to help the readers of Cancers, we have added to the text “cfDNA VAF also correlated with cfDNA concentration (r=0.484, p<0.05)” (page 9, line 271).

Reviewer 2 Report

Comments and Suggestions for Authors

This is a well written paper about a very interesing topic.

I just give you some suggestions:

-It would be preferable to differentiate the titles of paragraphs 3.1 and 3.2 that are now very similar

-Paragraph 3.2 lines 225-227: is there a correlation in LBCL between cfDNA level and bulky disease (a clinical parameter which is cited in the next section)?

-Lines 308-309: it would be better to exclude patients not treated with a standard anthracycline-based immunoCT (if they are included in the subgroup "other" Rituximab-containing regimens)

-Lines 336-344: please clarify how many patients are PET + at EOT and how many of them have a cfDNA increased vs reduced; please clarify how many patients are PET - at EOT and how many of them have a cfDNA increased vs reduced.

-Line 369: substitute "could be" with "was successfully isolated"

-Line 465: substitute "at the end of diagnosis" with "at the end of treatment"

-Supplementary Fig 2: PFS and OS are probably reported according to kinetics of ctDNA (< or > 2.5 log), not PET results

Comments on the Quality of English Language

The quality of English is good. I suggest revising the Discussion to make some sentences more fluent.

Author Response

RESPONSE TO REVIEWER 2:

Comments and Suggestions for Authors

This is a well written paper about a very interesting topic.

I just give you some suggestions:

We appreciate your time and effort in reviewing our manuscript. Enclosed are the specific responses and the appropriate modifications to the manuscript, which are highlighted in the resubmitted paper.

-It would be preferable to differentiate the titles of paragraphs 3.1 and 3.2 that are now very similar

Thank you very much for your observation. We totally agree and therefore we have changed both titles into the following ones:

3.1. Patients characteristics and levels of cfDNA according to lymphoma subtype.

3.2. cfDNA levels according to characteristics at presentation in the main lymphoma subtypes.

-Paragraph 3.2 lines 225-227: is there a correlation in LBCL between cfDNA level and bulky disease (a clinical parameter which is cited in the next section)?

Thank you very much for your comments that we find very interesting. Unfortunately, we do not have the information for Bulky measurement of the whole LBCL cohort (N: 88) in our historical medical records. However, we do have this information in the 49 LBCL cohort in which pre and post cfDNA were measured. According to the presence or absence of bulky disease in the LBCL cohort (N: 49) the results for cfDNA levels at diagnosis were as follows:

No bulky disease: median cfDNA 29.4ng/mL (IQR: 15.80-54.6).

Bulky disease: median cfDNA 44.4ng/mL (IQR: 20.3-57.8).

Unfortunately, no statistical significative differences were observed between the two groups (p value = 0.561) so we did not include this result in the paper.

However, and as mentioned in 3.3 section, we found interesting to show that those patients who had high tumor burden (defined as LBCL with advanced stage or those with localized stage but bulky disease) had a higher VAF than those with low tumor burden (Figure 6b).

-Lines 308-309: it would be better to exclude patients not treated with a standard anthracycline-based immunoCT (if they are included in the subgroup "other" Rituximab-containing regimens)

Thank you very much for pointing this out. The 4 patients considered to be treated with other rituximab containing regimens were treated with intensive CHOP-like chemotherapy with curative intention. Paliative regimens where exluded for this analysis. For this reason, and considering the small sample size, we support to maintain these 4 patients in the analysis.

-Lines 336-344: please clarify how many patients are PET + at EOT and how many of them have a cfDNA increased vs reduced; please clarify how many patients are PET - at EOT and how many of them have a cfDNA increased vs reduced.

According to the reviewers comments we have added the following information to the text (page 12, lines 339-343):

“At EOT, seven patients were PET+ (3 patients showed a reduction in cfDNA levels, 3 patients showed an increase in cfDNA, and 1 patient showed no significant cfDNA variation). Thirty-six patients were PET- at EOT (25 patients showed a reduction in cfDNA levels, 8 showed an increase in cfDNA, and 3 patients showed no significant cfDNA variation)”.

-Line 369: substitute "could be" with "was successfully isolated"

Thank you for pointing this out. We have changed this in the text.

-Line 465: substitute "at the end of diagnosis" with "at the end of treatment"

Thank you for pointing this out. We have changed this in the text.

-Supplementary Fig 2: PFS and OS are probably reported according to kinetics of ctDNA (< or > 2.5 log), not PET results

Thank you for pointing this out. We agree with you that the title is wrong. We propose change in it for the one: “PFS and OS based on the ctDNA kinetics (reduction of >2.5 log hGE/mL or < 2.5 log hGE/mL from diagnosis to the end of treatment)”.

 Comments on the Quality of English Language

The quality of English is good. I suggest revising the Discussion to make some sentences more fluent.

Thank you very much for your comments. We have reviewed the discussion and rewritten some sentences to make them more fluent. The new sentences are highlighted in the new version of the manuscript.

Page 14, line 398-403: “The relationship between circulating tumor DNA (ctDNA) levels and tumor burden or risk scores in FL remains incompletely defined. Nevertheless, recent studies have established correlations between baseline ctDNA, advanced disease stages, FLIPI score, and prognosis [23]. In cases of progressive FL with histological transformation, ctDNA has demonstrated the capability to detect gene mutations related to transformation that were not discernible in tumor biopsies [24].”

Page 14, line 413-416: “These aspects become particularly crucial in scenarios where accessing tumor tissue is challenging or in patients with a high risk of complications due to comorbidities. Additionally, ctDNA profiling can unveil tumor heterogeneity originating from various tumoral sites”.

Page 15, line 445-447: “The utility of baseline ctDNA concentration has been investigated for predicting treatment outcomes. Additionally, the measurement of ctDNA during treatment, at the conclusion of therapy, and throughout surveillance has also been explored.”
